# Physical and Physiological Profiles of Aerobic and Anaerobic Capacities in Young Basketball Players

**DOI:** 10.3390/ijerph17041409

**Published:** 2020-02-21

**Authors:** David Mancha-Triguero, Javier García-Rubio, Antonio Antúnez, Sergio J. Ibáñez

**Affiliations:** 1Grupo GOERD, Facultad de Ciencias del Deporte, Universidad de Extremadura, 10071 Cáceres, Spain; jagaru@unex.es (J.G.-R.); sibanez@unex.es (S.J.I.); 2Grupo GOERD, Facultad de Educación, Universidad de Extremadura, 06071 Badajoz, Spain; antunez@unex.es

**Keywords:** basketball, physical fitness, gender, field test, specific test, aerobic capacity, anaerobic capacity

## Abstract

Current trends in the analysis of the physical fitness of athletes are based on subjecting the athlete to requirements similar to those found in competition. Regarding physical fitness, a thorough study of the capacities that affect the development of team sports in different ages and gender is required since the demands are not equivalent. The objective of this paper was to characterize the physical-physiological demands of athletes in an aerobic and anaerobic test specific to basketball players, as well as the evolution of the variables according to age and gender. The research was carried out in 149 players from different training categories (*n* = 103 male; *n* = 46 female). The athletes performed two field tests that evaluated both aerobic capacity and lactic anaerobic capacity. Each athlete was equipped with an inertial device during the tests. Sixteen variables (equal in both tests) were analyzed. Three of them evaluated technical-tactical aspects, four variables of objective internal load, six kinematic variables of objective external load (two related to distance and four related to accelerometry) and three neuromuscular variables of objective external load. The obtained results show significant differences in the variables analyzed according to the age and gender of the athletes. They are mainly due to factors related to the anthropometric maturation and development inherent in age and have an impact on the efficiency and technical and tactical requirements of the tests carried out and, therefore, on the obtained results in the tests.

## 1. Introduction

Basketball is an invasion team sport, dynamic and intermittent in nature, formed by fast and short displacements, where changes in speed and direction are produced and where jumps are an integral part of the game’s demands [1]. These requirements require good physical conditioning, as it is essential to succeed in basketball [2]. However, the demands vary depending on the age, level and gender of the players [3]. For this, physical fitness can be evaluated through different components, including cardiorespiratory or muscular endurance [4].

Cardiorespiratory endurance can be classified into aerobic resistance and anaerobic resistance. Aerobic resistance is the capacity to maintain a stimulus for a prolonged period of time [5]. This causes the athlete to make adaptations to the effort (competition or training). After a while, energy production will be lower before the same stimulus, reaching a process of saving energy [6]. The importance of this capacity is due to the fact that the athlete must quickly recover from intense efforts in a short period of time in order to be able to chain a greater number of efforts during the competition.

On the other hand, anaerobic resistance is defined as the energy expenditure used by anaerobic metabolism (without the use of oxygen) that lasts less than 90 s, using an exhaustive effort [7]. In anaerobic metabolism, depending on time, energy is obtained through the adenosine triphosphate-phosphocreatine system (ATP-PCr), for actions lasting between 3 and 15 s during maximum effort [7]. From that moment on, the system responsible for energy production is anaerobic glycolysis, which can be maintained for the rest of the total effort [7]. The importance of this capacity in basketball is due to the fact that the best-classified teams in the competition are capable of performing a greater number of explosive actions. These high intensity actions are characterized by being of greater intensity and for longer than the worst classified teams [8].

Classified as a hybrid sport due to the importance of both metabolisms, Reference [9] identified that the players during the competition reached an average heart rate of 169 beats per minute and maximum heart rate of 192 beats per minute. In addition, the intensity of the competition analyzed through the % of the maximum heart rate during the game time (without counting breaks or stops) was 85% of their individual maximum heart rate. Along these lines, coinciding with the results obtained previously, Reference [10] confirmed that 19.3% ± 3.5% of the game time, the player’s heart rate exceeds 95% of its maximum value, whereas during 56.0% ± 6.3% of the game time, the heart rate is in the range of 85–95% of the maximum heart rate.

When practicing a sport, the load supported by the athlete can be assessed through different instruments or methodology [11]. This assessment can be carried out taking into account the athlete´s internal or external load. In addition, this assessment can be objective or subjective type (depending on the instruments and resources). At present, one of the variables most used to know the athlete´s fatigue is to know the distance travelled. During a basketball match, this variable (like any other) can be influenced by different contextual variables (age, gender or level) with results between 6000 and 7500 m [12,13]. In addition, regarding the anaerobic component of the competition, around 20% of the number of total actions carried out is classified as high-intensity actions [10]. Along these lines, coinciding with the above, Reference [14] describe that each player performs approximately 1000 short actions that change every 2 s.

The importance of quantifying the demands supported by athletes during the competition affects the training processes since, during training, athletes should be prepared to achieve the best possible results related to sports performance. To do this, a reliable technique to know the physical-physiological state of the athlete is to carry out physical fitness tests. These tests can be grouped according to the type of test and the specificity. The type of test refers to the place where they are performed. They can be field or laboratory. Specificity refers to the relationship with the sport to be assessed they can be specific or non-specific [15].

The analysis of the internal and external load by means of a test of aerobic and lactic anaerobic capacities is not a common practice in players at training stages. Currently, it is normally used the information that comes from high-level teams [16] or national teams [17], evaluated through laboratory or generic tests regardless of the sport practiced [18]. For this reason, it is believed that the analysis of physical fitness in young athletes is relevant since occasionally the values obtained by the high-level teams are adapted and the principle of specificity and individualization of the athlete is omitted [19]. Thus, the monitoring of physical fitness in each training period and gender provides the coach with relevant information when planning a competition, existing then a positive relationship between a better physical fitness and a better performance of the athlete in the competition [20]. Once the literature has been revised, there is no objective knowledge of how aerobic and anaerobic capacities evolve in the different educational stages of basketball. Therefore, the objectives of this research were to characterize the physical-physiological demands belonging to aerobic and anaerobic capacity in different educational stages of basketball and in male and female players through specific field basketball tests. Besides, it is intended to describe the differences between genders according to age.

The hypothesis of this manuscript is that aerobic and anaerobic capacities will obtain an increase in the analyzed values as the athlete develops. For this reason, older players will have better results in these capacities than younger players. In addition, differences between players of the same age, but of different genders, will be confirmed. Male players are the ones who will obtain the best results in these physical fitness tests.

## 2. Materials and Methods

### 2.1. Design

This research falls within the empirical studies that follow an associative strategy, which seek to examine the differences between groups and relationships between tests, through Comparative Studies of a cross-sectional and evolutionary type [21], in order to characterize the performance of basketball players in different genders and age through physical fitness tests.

### 2.2. Participants

One hundred and forty nine male and female players that belong to teams of different ages (U´14, U´16 and U´18) and that belong to the same club and participate in the national championship (U´14 male: *n* = 33, Weight = 62.20 kg, Height = 1.72 m, BMI = 20.78; U´14 female: *n* = 12, Weight = 53 kg, Height = 1.60 m, BMI = 21.875; U´16 male: *n* = 31, Weight = 76.81 kg, Height = 1.87 m, BMI = 21.91; U´16 female: *n* = 12, Weight = 60.39 kg, Height = 1.64 m, BMI = 22.34; U´18 male: *n* = 39, Weight = 85.23 kg, Height = 1.95 m, BMI = 22.41; U´18 female: *n* = 22, Weight = 57.3 kg, Height = 1.68 m, BMI = 20.59) were evaluated. In order to be part of the research, athletes had to carry out both tests so that the sample was the same and the choice of the subjects was not a confounding variable. Both the coaching staff and the players were previously informed of the details of the research and its possible risks and benefits, the participation of the athletes was voluntary. For this, they were asked for their approval about the participation through informed consent. In underage players, the consent was signed by their parents or legal guardians. The study was developed based on the ethical provisions of the Declaration of Helsinki (2013), approved by the Bioethics Committee of the University (registration number 233/2019).

### 2.3. Procedure

First, we made contact with the club and the coaches to inform them about the project. Once the proposal was accepted, an informed consent was made for the parents with relevant information about the research. Secondly, the calendars with the competitions of the teams were analyzed and the moments of absence of competition were selected in order to carry out the physical fitness tests so that the athletes were in the best physical conditions possible and the generated fatigue did not affect subsequent competitions. After the data collection of each team, a dossier was made for the coach with the information of the tests in order to have more knowledge about the physical fitness of the players. To carry out the tests, the protocol described in the Specific Battery Fitness Test (SBAFIT) test battery [15] was used. The tests were carried out on two different days (The first day the aerobic capacity test. On the second day, the lactic anaerobic capacity test was performed) separated by at least 72 h of recovery so that the players made a complete recovery and the results were more reliable. Finally, all the participants in the study conducted 2 training sessions with the material used in the measuring to practice the tests to be evaluated in order to have the first contact so that ignorance or discomfort were not confounding variables.

### 2.4. Instruments and Equipment

For the recording of the technical-tactical variables, a video camera JVC model GY-HM70U with a sampling rate of 300 fps (resolution 720 × 480). In addition, a registration sheet was used to account for the shots to the basket and the score or error sequence. For the registration of objective internal load variables, each athlete was equipped with a heart rate band from GARMIN^®^ (Olathe, KS, USA) and for the recording of the Objective External Load Kinematics Variables related to Distance, the Objective External Load Kinematics Variables related to Accelerometry and the Objective External Load Neuromuscular Variables, each player was equipped with a WIMU^®^ inertial device from RealTrack Systems (Almería, Spain), which was fixed using a harness anatomically adjusted to each player. After registration, the data was analyzed using the SPRO^®^ software from RealTrack Systems (Almería, Spain). The WIMU inertial device has a sampling frequency of 18 Hz.

To assess the aerobic capacity and lactic anaerobic capacity of athletes, two validated tests were carried out, the SIG/AER Aerobic Test [22] which consists of a twelve-minute test with the objective of making the greatest number of possible laps in a circuit (1 circuit = 12 fractions) in which the individual performs different technical-tactical sport actions such as different movements with and without the ball, shots to the basket and rebound or defense action. These actions are structured in a sequential order similar to the one which can happen in a match and on the court [5]. The SIG/ANA Test [23] consists of a five-minute performance test that is carried out intermittently (one minute of activity, one minute of passive recovery) in order to make as many laps as possible to a circuit (1 circuit = 4 fractions) in which the subject performs different technical-tactical sport actions such as different movements with and without the ball, shots to the basket and rebound or defense action, among others. The test seeks the ability to repeat a maximum activity with incomplete recoveries. These actions are structured in a sequential order similar to the one which can happen in a match and on the court. This test looks for similarity to the SIG/AER test in which the structure is modified to be able to evaluate the selected capacity [5].

### 2.5. Variables

For this research, age (U´14, U´16 and U´18) and gender (male and female) were defined as independent variables. For the assessment of the physical fitness of athletes, the following variables were analyzed, which are divided into five groups according to the type of demands [24]: i) Technical-Tactical Variables, ii) Objective Internal Load Variables, iii) Objective External Load Kinematic Variables related to Distance, iv) Objective External Load Kinematic Variables related to Accelerometry, v) Objective External Load Neuromuscular Variables.

i) Technical-Tactical Variables were analyzed by means of observational methodology the technical tactical gestures made during each test.

i.1) Shots: It is the total number of shots the player makes during the duration of the test.i.2) Scores: It is the number of scored shots.i.3) Efficacy (%): It is the value (expressed in %) calculated of the product between Scores and Number of Shots.

There is a document that provides guidance values on these variables based on the age and gender of the players [5].

ii) Objective Internal Load Variables were evaluated through the Heart Rate (HR). It is an individual indicator of the athlete´s demands on a task or training. Within this variable, the following related parameters are analyzed.

ii.1) Heart Rate Maximum (HR Max): Maximum value of beats per minute reached by the athlete during the test.ii.2) Heart Rate Medium (HR Med): Average value of beats per minute during the test.ii.3) % Heart Rate Maximum (% HR Max): It is an indicator of the intensity of the physical-physiological effort of the athlete during the test. This value is calculated taking into account the HR Max.ii.4) Heart Rate Recovery (HR Rec): Value of beats per minute after two minutes of the end of the test. The athlete must do a passive recovery at the end of the tests [15].

iii) Objective External Load Kinematics Variables related to Distance. They analyze the external load borne by players during the test time and their movements.

iii.1) Part of Circuit: Number of circuit fragments made by the player during the duration of the test. In the aerobic test, each circuit is made up of 12 fractions, whereas in the anaerobic test, each circuit is made up of 4 fractions.iii.2) Distance (m): The number of meters travelled during the test. In the aerobic test, each fraction has an approximate distance of 15 m and in the anaerobic test, each fraction has an approximate distance of 7.5 m [5].

There are some guidance tables on the level of athletes according to age and gender [5].

iv) Objective External Load Kinematics Variables related to Accelerometry. They register the external load made by players in relation to the execution time and their movements.

iv.1) Accelerations: Positive increase in speed made during the game, total and per minute.iv.2) Decelerations: Negative increase in speed made during the game, total and per minute.

v) Objective External Load Neuromuscular Variables. They analyze the external load that the player receives in relation to the gravitational force. Two variables are recorded:

v.1) Impacts: They are measured through the force that the musculoskeletal structures support in relation to gravitational force (g-force).v.2) PlayerLoad: It is a vectorial magnitude derived from triaxial accelerometry data that quantifies the movement at a high resolution. Accelerations and decelerations are used to construct a cumulative measure of the rate of change in acceleration. A cumulative measure (PL) and a measure of intensity (PL · min^−1^) are used and can then indicate the stress rate at which players subject their body for a certain period of time. As a unit load, it has a moderate-high degree of reliability and validity [25,26].

The variables Accelerations, Decelerations and PlayerLoad were normalized per minute with analysis purposes.

### 2.6. Statistical Analysis

First, a descriptive analysis of the quantitative variables (Mean and Standard Deviation) was carried out. Secondly, an exploratory analysis was performed using the assumption of criteria tests [27], finding a normal distribution of data, so parametric tests were carried out for the hypothesis testing. An analysis was conducted to compare performance between categories using the T-test for Independent Samples to identify significant differences related to age and gender. Finally, the effect size was calculated using Cohen´s d, being classified as low effect (0–0.2), small effect (0.2–0.6), medium effect (0.6–1.2), large effect (1.2–2.0) and very large effect (>2.0) [28]. The software used was SPSS 23.0 (SPSS Inc., Chicago, IL, USA). The significance was established at the value of *p* < 0.05 [29].

For the graphic presentation of the data (Figure 1 and Figure 2) the normalization of the results was carried out through *Z Scores.* The purpose of the Z-Score is to standardize a value so that it represents the number of standard deviation the value is above the mean [30]. The results were presented in profiles based on the age and gender of the sample of athletes.

## 3. Results

Figure 1 (Appendix A) shows the results of the aerobic test grouped by gender and age. It can be observed that the analyzed variables show differences according to age and gender. In the case of males, the U´16 sample obtains the highest results in most of the variables compared to U´14 and U´18 samples, whereas in the female category, the U´16 sample obtains worse results in the majority of variables than U´14 or U´18 samples.

Figure 2 (Appendix A
Appendix A) shows the results of the anaerobic test grouped by gender and age. It can be observed that the analyzed variables show differences according to age and gender. In U´16 and U´18 samples in the male category, the obtained results are similar mainly in the technical-tactical variables and in the objective internal load variables, whereas in the female category of the U´16 sample, as in the aerobic test, they obtain worse results in most variables than in the samples U´14 or U´18.

Table 1 shows the results of the differences of the variables analyzed according to age and gender. The obtained results show that the lowest values of the Effect Size occur in male players. However, in the female category, the values in Cohen´s d (Effect Size) in the categories U´14–U´18 and U´16–U´18 are the highest.

Table 2 shows the differences in each test according to the analyzed variables between male and female players. The variables with significant differences show that there are differences in the value obtained between male and female athletes. In addition, the Effect size of each variable is provided.

As shown in Table 2, there are significant differences in some variables depending on gender. The age that shows more differences between genders is the U’18 category, while the U’14 category is the one that provides less significant differences. The differences between genders depending on the test show that there is a greater number of significant differences in the variables belonging to the anaerobic test than to the aerobic test.

## 4. Discussion

The objective of this study was to characterize the physical-physiological demands pertaining to the aerobic and anaerobic capacities in different basketball ages and among male and female basketball players. In addition, it was intended to identify gender differences based on age. The main results show that there are significant differences between players according to age and gender.

### 4.1. Aerobic Capacity

All groups of variables analyzed show significant differences according to age and gender. The Shots variable is directly related to Parts of Circuits and Distance (m.), since a shot is made every 12 Parts of Circuits in the aerobic capacity test. Therefore, the higher the number of parts of circuits, the higher the number of shots. Related to the latter, Reference [13] describe that a player travels between 6000 and 7500 m during a match depending on the level, age or specific position (linked to anthropometric factors). In this research, male players travel around 2000 m in a 12 min test, while in the female gender, the distance travelled during the test is approximately 1800 m. If the distance traveled during the test is extrapolated to the duration of a match, the results coincide with those existing in the literature [13]. Along these lines, it can be affirmed that the aerobic level of male players is Medium or Medium-High, while the level of female players is classified as Medium or Medium-Low [5]. Regarding gender differences, no significant differences between male and female players of the same age are identified in the aerobic test.

However, in Efficacy, there are differences between players U’14 vs. U’16 and U’14 vs. U’18 in both genders. The percentage of scoring can be influenced by different aspects such as the technical level or fatigue [31]. Along these lines, Reference [5] define the values of efficacy that players should have according to age and the capacity to be evaluated during the test. In the aerobic test, the male athletes in the sample obtain the value of Fair or Fair-Good, whereas the female athletes obtain Bad or Bad-Fair values. A lower Efficiency in younger athletes may be due to the stage they are in (puberty), which is characterized by a growth in body segments that can be linked to a motor decompensation and a lower technical-tactical level. In this investigation, the results show that the male athletes who have greater value in objective internal load variables (HR) coincide with the athletes who have lower Efficacy during the test. The better physical fitness directly affects the technical-tactical efficiency. As for female players, the results do not match those found in the male gender and may be due to the technical level of the players. There are significant differences between male and female players in the U’16 category in the variable Scores and in U’18 in the variable Efficacy. These differences may be due to the fact that male players support lower demands during the development of the tests before the same stimulus than female players. A lower demand may cause less fatigue what has an impact on the shot technique and, therefore, on the result [31].

The obtained results provide relevant information about male athletes as they present different demands from female athletes during the aerobic test. In acceleration and deceleration variables as well as, in the variables relativized per minute, U16 male players obtain the highest values, while U18 players are the ones that obtain them in the female gender. Along the lines above, Reference [8] define that explosive or high-intensity actions are a predictor of performance, being the winning team the one that performs the greatest number of actions since basketball is a high-intensity sport with continuous changes of direction [12]. In the aerobic test, there are significant differences between male and female players in the categories U’14 and U’18 in all the variables of this type analyzed.

Related to the previous paragraph, both accelerations and neuromuscular variables have an impact on the athlete’s load. In this line, there are significant differences between male and female players in U’16 and U’18 categories in Impacts and PlayerLoad/minute variables, while in PlayerLoad there are only significant differences between U’18 male and female players. The Impacts variable obtains its values because of the product of the G forces that the athlete suffers. Therefore, athletes with greater anthropometry (male players) obtain greater value in the variable. In PlayerLoad and PlayerLoad/minute variables, the value is calculated as a result of the accelerometry that athletes support based on their movements. Although female athletes give a more physical performance during the competition, the final physical performance is lower than that of male players [32] due to factors related to physical efficiency.

The causes of the differences mentioned in this section may be due to the relationship with other variables analyzed such as the distance travelled since the longer the distance, the higher the number of intervallic actions the athlete will perform because the circuit made is repeated systematically. Furthermore, the literature states that these differences exist due to factors related to maturation and growth since the relative VO2max (depending on weight) is an optimal indicator of the physiological capacities of athletes [17]. These differences are mainly due to the fact that male players perform more intense actions during the season than female players [33]. The differences between genders can be the result of maturation (linked to aspects related to strength) or of the morphology of athletes (anthropometry).

Finally, there are many studies highlight the importance of a well-developed aerobic capacity in basketball players for the recovery of many high-intensity actions that are accumulated when practicing [33], with incomplete breaks [15]. In addition, the test carried out confronts the athletes with similar demands that they find during the competition. Therefore, during the competition, the athlete is in similar values to those obtained in the test, close to 90% of the HR Max [34]. These differences affect the production of VO2max as it is higher depending on age because it is linked to greater development and body size [35]. In % HR Max, all players are at values higher than 75% of the HR Max during the aerobic test [36].

### 4.2. Anaerobic Capacity

As in the aerobic capacity test, all groups of variables analyzed show significant differences according to age and gender. In this respect, the Shots variable is directly related to Parts of Circuits and Distance (m.), since a shot is made every4 Parts of Circuits. Therefore, the higher the number of parts of circuits, the higher the number of shots. As in the aerobic test, this investigation shows differences between U’14 vs. U’16 players and U’14 vs. U’18 in both genders. The literature states that these differences are due to factors related to maturation and growth [37]. In this case, the analyzed sample of male players is at a Medium level, while female players are at a Medium and Medium-High level [5]. Furthermore, there are significant differences between genders in U’14, U’16 and U’18 categories both in the Parts of Circuits variable and in the Distance variable. These differences confirm the results found in the literature where men travel a longer distance at high intensity, while women carry out a greater volume of demands [32,33,34,35,36,37].

Regarding the Efficacy variable, as in the aerobic test, there are differences between players of different categories and in both genders. The percentage of scoring can be influenced by different aspects such as the technical level or fatigue [31]. Along these lines, Reference [5] define the values of efficacy that players U’14 and U’16 male and female players obtain values higher than Very Good, while U’18 players of each gender obtain a Good value. In relation to the Heart Rate variables (HR), as in the aerobic test, the male players who obtain the highest value in the HR Variables, are the players who obtain the lowest value of effectiveness in the launch of the test. However, in female players, this does not happen and there is no relationship and may be due to factors related to the technical level.

The results of Kinematic Variables Related to Accelerometry are in the previous line, as in the aerobic capacity test, Reference [8] define that explosive or high-intensity actions are a predictor of performance, being the winning team the one that performs the greatest number of actions since basketball is a high-intensity sport with continuous changes of direction [12]. As for the results obtained, the ones obtained in acceleration and deceleration as well as in those relativized per minute, show differences that may be due to the distance traveled during the test. Furthermore, there are significant differences between genders in categories U’14, U’16 and U’18 in Accelerations and Accelerations/minute, while in Decelerations and Decelerations/minute there are only significant differences between male and female players in U’16 and U’18 categories. These differences are mainly due to the fact that male players perform more intense actions during the season than female players [32]. The differences between genders can be the result of maturation (linked to aspects related to strength) or of the morphology of athletes (anthropometry).

Related to the previous paragraph, the results in neuromuscular variables show significant differences in the Impacts variable between all categories. These differences are largely due to the difference between anthropometric aspects [20] of athletes due to their development or maturation [38]. Players who support more impacts during a competition or physical fitness test are more likely to be injured during the season [20]. For this reason, as in the aerobic capacity, neuromuscular variables and those related to accelerometry, are a clear indicator of neuromuscular fatigue that must be taken into account during the competition or training, in order not to cause the athlete a period of overtraining that causes a temporary performance deficit [39]. In addition, the Impacts variable is related to the athlete’s anthropometry.

Although female athletes give a more physical performance during the competition due to the volume of demands, the male players get lower volume. However, the intensity of the male players is higher, this key difference being in the final performance of the competition [40]. For this, the importance of the analysis of these two capacities is due to the fact that anaerobic capacity predicts performance better than aerobic capacity [41]. On the contrary, Reference [42] showed a positive correlation between aerobic aptitude and the game level in both male and female gender. These differences found in the literature show the relevance of carrying out physical fitness tests in athletes in both capacities.

The limitations found in this investigation were that the athletes analyzed were part of teams that participated in the National Championship in their corresponding categories. This sample does not provide a generic knowledge of the categories analyzed since not all teams are part of this championship (only the best ones from different geographical regions). Besides, within the championship there are many differences between the teams that play the final stage of the championship and the teams that only play the group stage. Therefore, it would be interesting in further studies to expand the sample to teams from different geographical areas and with different results in national championships.

## 5. Conclusions

The analysis of the physical fitness of basketball players through specific field tests measured with microtechnology instruments provides objective and reliable knowledge to the coach and the physical coach. For this, physical fitness assessments are carried out through physical fitness tests. The choice of the test is important and affects the final result. Tests that are specific to the sport to be evaluated and that are carried on the court or training place must be selected in order not to affect the athlete nor produce unreliable results. Subjecting athletes to laboratory tests sometimes means exposing them to stress what can have an impact on the result of the test and, therefore, on the final result in the competition.

Aerobic and anaerobic physical demands in male players increase with age and there is improvement in technical-tactical performance. The obtained results progressively increase with age in the three categories, showing the influence of maturational development and sports experience. Aerobic and anaerobic physical demands in female players progressively increase with age and stabilize from the age of 16 reaching a plateau, their growth being more gradual.

The differences in the three categories between genders are verified, both at the level of Objective Internal Load and Objective External Load variables as well as Technical-Tactical variables. It is proved that maturational development has already occurred in these categories, differentiating the game between males and females.

It is recommended that from the age of pubertal changes the training between male and female players is differentiated because they present different physical and technical-tactical manifestations, something that in previous ages does not happen, and they are able to train together.

U’16 male players obtain better results in the tests carried out than the rest of the players. This evolution is not the result of the maturation process in which they are and could be due to the training processes carried out by each team, in which they train more efficiently and optimally than the rest of the athletes.

## Figures and Tables

**Figure 1 ijerph-17-01409-f001:**
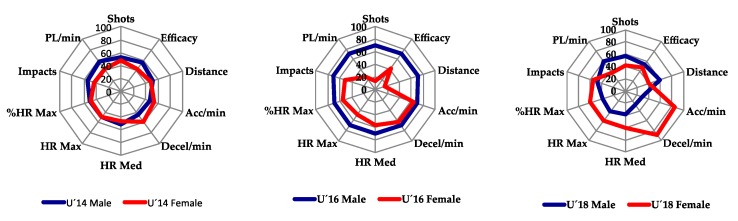
Standardized results of the aerobic test grouped by gender and age.

**Figure 2 ijerph-17-01409-f002:**

Standardized results of the anaerobic test grouped by gender and age.

**Table 1 ijerph-17-01409-t001:** Inferential analysis and Effect Size of the variables analyzed according to age and gender.

			Aerobic Capacity				Anaerobic Capacity			
			*MALE*		*FEMALE*		*MALE*		*FEMALE*	
			*sig.*	*ES*	*sig.*	*ES*	*Sig.*	*ES*	*sig.*	*ES*
Technical-tactical Variables	Shots	U´14–U´16	0.000 *	−1.524	0.099	0.682	0.000 *	−0.662	0.099	0.585
		U´14–U´18	0.000 *	−1.384	0.000 *	−1.243	0.000 *	−0.658	0.000 *	−0.618
		U´16–U´18	0.287	0.312	0.000 *	−2.386	0.287	0.049	0.000 *	−1.38
	Scores	U´14–U´16	0.000 *	−1.226	0.843	−0.816	0.000 *	−1.617	0.843	−0.561
		U´14–U´18	0.000 *	−0.967	0.008 *	−0.839	0.000 *	−1.341	0.008 *	−0.998
		U´16–U´18	0.698	0.197	0.292	−0.47	0.698	0.12	0.292	−0.504
	Efficacy	U´14–U´16	0.000 *	−0.89	0.306	−0.991	0.000 *	0.049	0.306	−0.345
		U´14–U´18	0.000 *	−0.711	0.04 *	−0.7	0.000 *	−0.054	0.04 *	0.563
		U´16–U´18	1.000	0.136	1.000	−0.128	1.000	−0.111	1.000	0.93
Objective External Load Kinematics V. related to Distance	Parts of Circ.	U´14–U´16	0.000 *	−1.176	0.017 *	0.876	0.000 *	−0.515	0.017 *	0.355
		U´14–U´18	0.000 *	−1.035	0.181	−0.535	0.000 *	−0.483	0.181	−0.889
		U´16–U´18	0.287	0.321	0.000 *	−1.915	0.287	0.066	0.000 *	−1.329
	Distance (m)	U´14–U´16	0.000 *	−1.176	0.017 *	0.875	0.000 *	−0.515	0.017 *	0.355
		U´14–U´18	0.000 *	−1.034	0.181	−0.536	0.000 *	−0.483	0.181	−0.888
		U´16–U´18	0.287	0.321	0.000 *	−1.915	0.287	0.065	0.000 *	−1.329
Objective External Load Kinematics V. related to Accelerometry	Acc.	U´14–U´16	0.978	−0.172	0.000 *	−1.257	0.978	−0.126	0.000 *	−0.499
		U´14–U´18	1.000	−0.011	0.000 *	−2.453	1.000	1.447	0.000 *	−1.208
		U´16–U´18	1.000	0.178	0.18	−0.633	1.000	1.387	0.18	−0.918
	Decel	U´14–U´16	0.229	−0.307	0.000 *	−1.257	0.229	−0.806	0.000 *	−1.247
		U´14–U´18	1.000	−0.142	0.000 *	−3,069	1.000	0.781	0.000 *	−1.497
		U´16–U´18	1.000	0.186	0.002 *	−1.093	1.000	1.15	0.002 *	−0.735
	Acc/min	U´14–U´16	0.751	−0.2	0.000 *	−1.264	0.751	−0.114	0.000 *	−0.412
		U´14–U´18	1.000	−0.045	0.000 *	−2.438	1.000	1.342	0.000 *	−1.216
		U´16–U´18	1.000	0.174	0.188	−0.626	1.000	1.303	0.188	−1.01
	Decel/min	U´14–U´16	0.163	−0.332	0.000 *	−1.264	0.163	−0.832	0.000 *	−1.105
		U´14–U´18	0.881	−0.174	0.000 *	−3.049	0.881	0.704	0.000 *	−1.503
		U´16–U´18	1.000	0.18	0.002 *	−1.087	1.000	1.068	0.002 *	−0.821
Objective Internal Load V.	HR Med	U´14–U´16	0.021 *	0.486	1.000	−0.266	0.021 *	0.648	1.000	−0.111
		U´14–U´18	0.000 *	1.122	1.000	−0.247	0.000 *	1.758	1.000	0.488
		U´16–U´18	0.001 *	0.636	1.000	−0.07	0.001 *	0.58	1.000	0.829
	HR Max	U´14–U´16	0.241	0.337	0.496	0.514	0.241	0.837	0.496	1.708
		U´14–U´18	0.000 *	0.775	1.000	−0.161	0.000 *	1.868	1.000	1.434
		U´16–U´18	0.023 *	0.455	0.103	−0.621	0.023 *	0.591	0.103	0.405
	HR Rest	U´14–U´16	0.013 *	−0.575	0.056	1.148	0.013 *	1.424	0.056	−0.953
		U´14–U´18	0.000 *	0.774	0.003 *	0.925	0.000 *	2.477	0.003 *	0.764
		U´16–U´18	0.000 *	1.337	1.000	0.116	0.000 *	0.906	1.000	1.792
	% HR Max	U´14–U´16	0.059	0.453	1.000	0.171	0.059	−0.203	1.000	−0.57
		U´14–U´18	0.000 *	1.053	1.000	−0.035	0.000 *	0.403	1.000	0.042
		U´16–U´18	0.002 *	0.636	1.000	−0.143	0.002 *	0.569	1.000	0.179
Objective External Load Neuromuscular V.	Impacts	U´14–U´16	0.179	0.359	1.000	0.768	0.179	−0.599	1.000	1.037
		U´14–U´18	1.000	−0.081	0.002 *	−0.952	1.000	−1.152	0.002 *	0.572
		U´16–U´18	0.057	−0.397	0.000 *	−1.275	0.057	−0.407	0.000 *	−0.971
	PlayerLoad	U´14–U´16	0.000 *	−0.761	0.005 *	0.974	0.000 *	−0.213	0.005 *	1.561
		U´14–U´18	0.000 *	−0.941	0.185	−0.616	0.000 *	0.439	0.185	2.165
		U´16–U´18	1.000	0.034	0.000 *	−1.785	1.000	0.578	0.000 *	0.086
	PlayerLoad/min	U´14–U´16	0.000 *	−0.767	0.006 *	0.976	0.000 *	−0.213	0.006 *	1.56
		U´14–U´18	0.007 *	−0.656	0.168	−0.625	0.007 *	0.439	0.168	2.164
		U´16–U´18	0.323	0.034	0.000 *	−1.792	0.323	0.578	0.000 *	0.083

Parts of Circ: Parts of Circuits; Acc: Accelerations; Decel: Decelerations; HR Med: Heart Rate Medium; HR Max: Heart Rate Maximum; HR Rest: Heart Rate Recovery; PlayerLoad/min: PlayerLoad/ minute; sig: *p*-value; *: *p* < 0.05; ES: Effect Size.

**Table 2 ijerph-17-01409-t002:** Analysis of the differences per gender of athletes according to age.

			Aerobic Capacity		Anaerobic Capacity	
			*Sig.*	*ES*	*Sig.*	*ES*
Technical-tactical V.	Shots	U´14	0.606	0.242	0.002 *	−0.499
		U´16	0.167	2.571	0.22	0.645
		U´18	0.819	0.679	0.001 *	−0.25
	Scores	U´14	0.103	0.597	0.001 *	−0.654
		U´16	0.002 *	1.255	0.798	0.78
		U´18	0.075	0.402	0.086	0.041
	Efficacy	U´14	0.279	0.536	0.126	−0.475
		U´16	0.201	0.679	0.091	−0.994
		U´18	0.013 *	0.307	0.264	0.034
Objective External Load Kinematics V. related to Distance	Parts of Circuits	U´14	0.78	0.168	0.003 *	−0.143
		U´16	0.064	2.420	0.07 *	0.743
		U´18	0.436	0.914	0.000 *	−0.132
	Distance (m.)	U´14	0.78	0.169	0.003 *	−0.144
		U´16	0.064	2.420	0.07 *	0.743
		U´18	0.436	0.914	0.000 *	−0.132
Objective External Load Kinematics V. related to Accelerometry	Acc.	U´14	0.03 *	−0.294	0.004 *	0.865
		U´16	0.403	−1.480	0.000 *	0.364
		U´18	0.000 *	−2.167	0.000 *	−1.626
	Decel	U´14	0.001 *	−0.440	0.596	−0.172
		U´16	0.515	−1.318	0.01 *	−0.337
		U´18	0.000 *	−2.35	0.000 *	−1.555
	Acc/Min	U´14	0.024 *	−0.333	0.021 *	0.868
		U´16	0.436	−1.503	0.000 *	0.476
		U´18	0.000 *	−2.163	0.000 *	−1.518
	Decel/Min	U´14	0.001 *	−0.476	0.735	−0.161
		U´16	0.502	−1.339	0.016	−0.186
		U´18	0.000 *	−2.351	0.000 *	−1.45
Objective Internal Load V.	HR Med	U´14	0.008 *	0.202	0.000 *	0.352
		U´16	0.307	−0.494	0.015 *	0.458
		U´18	0.815	−1.155	0.472	0.987
	HR Max	U´14	0.677	−0.009	0.044 *	0.217
		U´16	0.037 *	0.220	0.005 *	0.56
		U´18	0.794	−0.859	0.878	0.79
	HR Rest	U´14	0.289	−0.611	0.003 *	0.487
		U´16	0.106	1.041	0.154	0.98
		U´18	0.804	−0.389	0.543	0.57
	% HR Max	U´14	0.006 *	0.034	0.04 *	0.231
		U´16	0.309	−0.346	0.615	0.345
		U´18	0.734	−1.084	0.002 *	0.974
Objective External Load Neuromuscular V.	Impacts	U´14	0.162	−0.049	0.000 *	−1.511
		U´16	0.011 *	−0.021	0.001 *	0.361
		U´18	0.003 *	−0.951	0.000 *	0.413
	PlayerLoad	U´14	0.447	0.668	0.21	0.062
		U´16	0.062	1.802	0.000 *	1.175
		U´18	0.003 *	1.228	0.000 *	1.136
	PlayerLoad/Min	U´14	0.525	0.680	0.209	0.063
		U´16	0.041 *	1.776	0.000 *	1.175
		U´18	0.01 *	0.896	0.000 *	1.135

Parts of Circ: Parts of Circuits; Acc: Accelerations; Decel: Decelerations; HR Med: Heart Rate Medium; HR Max: Heart Rate Maximum; HR Rest: Heart Rate Recovery; PlayerLoad/min: PlayerLoad/ minute; sig: *p*-value; *: *p* < 0.05; ES: Effect Size.

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
