# Peer review of "Physical and Physiological Profiles of Aerobic and Anaerobic Capacities in Young Basketball Players"

_ijerph, 2020, doi:10.3390/ijerph17041409_

Round 1
Reviewer 1 Report
I would like to thank you for submitting and give me the opportunity to review this interesting work about the characterisation of aerobic and anaerobic capacities in young basketball players. I hope my comments will help to improve the quality of the manuscript in some way.
In general, the manuscript is well structured and clear. However, results and discussion sections are a bit long and hard to follow because of the large number of variables studied in this work.
English in the manuscript is mostly good, but some parts of the text should be reviewed since there are some very long sentences with a muddle structure, especially in the introduction section. The editorial office will be able to assist you with the few corrections necessary when required.
Below I will make some specific comments on each section that should be carefully reviewed.
Introduction
The introduction is a bit general; it should be more focus on the aim of the work instead of in the general aspect of the sport physiology. In this sense, authors aim to characterize physical and physiological demands on different ages and genders in basketball, but there is nothing related to age or gender in basketball in the introduction that makes the reader understand why It is important to do this research.
In addition, it is necessary to add hypothesis to know what the authors expected to found with this research.
Methods
From my view point “variables” section should be after procedure section since the reader could understand when and why each variable has been measured. In this sense, I think that the best way to order the methods section is Design – Participants – Procedures - Instruments – Variables – Statistical Analysis.
A characterisation (age, height, weight…) of each group is needed in participants section.
Results
The presentation of the results is a bit dense; there are 4 very large tables with many numbers. Maybe the presentation of some results in graphs could help to show and interpret them more easily.
Discussion
As I point at the beginning of the review, discussion section is a bit long and at some point is hard to follow the text. Authors should consider reorganizing this section dividing results in two parts grouping some variables. They can do it in two different ways: age and gender or aerobic and anaerobic capacity.
The second paragraph (Page 11 Lines 272-281) should be in the introduction section because it is a review of the state of art in relation with young athletes.
In many parts of the discussion, authors explained the differences of some of the variables because of the level of maturation or the anthropometry, but there is not a characterisation of the sample in the methods section that justify the differences of the participants of each group.
Limitations and further research should be added at the end of the discussion section.
Author Response
Dear Editor,
We have carefully considered all reviewers' considerations of the paper (IJERPH-709605). Please find enclosed our detailed answers to reviewers' queries. Authors declare that the manuscript is original and has not been considered for publication elsewhere. Additionally, the authors had approved the paper for release and are in agreement with its content.
(x) I would not like to sign my review report.
() I would like to sign my review report English language and style.
() Extensive editing of English language and style required.
(x) Moderate English changes required.
We welcome your comment in the form of a proposal. The manuscript has been revised and some changes have been made with regard to the English language.
() English language and style are fine/minor spell check required.
() I don't feel qualified to judge about the English language and style.
|
Yes |
Can be improved |
Must be improved |
Not applicable |
|
|
Does the introduction provide sufficient background and include all relevant references? |
( ) |
(x) |
( ) |
( ) |
|
Is the research design appropriate? |
(x) |
( ) |
( ) |
( ) |
|
Are the methods adequately described? |
( ) |
(x) |
( ) |
( ) |
|
Are the results clearly presented? |
( ) |
(x) |
( ) |
( ) |
|
Are the conclusions supported by the results? |
(x) |
( ) |
( ) |
( ) |
Comments and Suggestions for Authors
Does the introduction provide sufficient background and include all relevant references?
The authors appreciate your comment in the form of a proposal with the aim of improving the quality of the manuscript. The Introduction has been improved in order to provide the reader with more details and information.
Are the methods adequately described?
The authors are grateful for your comment in the form of a proposal with the aim of improving the quality of the manuscript. The methodology section has been modified following the proposals as requested by the reviewers.
Are the results clearly presented?
The authors are grateful for the comment in the form of a proposal with the aim of improving the manuscript. Regarding this section, the results have been modified, with tables 1 and 2 becoming part of the Supplementary Materials and replacing this information in the final document with figures that facilitate reading. The data represented in the figures have been normalised through Z Scores. This choice is due to the fact that it becomes more interesting and is better represented, even though there are variables with different ranges of values. Such technique is commonly used to make graphs when there are many differences between the values of different variables.
I would like to thank you for submitting and give me the opportunity to review this interesting work about the characterisation of aerobic and anaerobic capacities in young basketball players. I hope my comments will help to improve the quality of the manuscript in some way. In general, the manuscript is well structured and clear. However, results and discussion sections are a bit long and hard to follow because of the large number of variables studied in this work.
The authors appreciate your comment and proposals. We have made changes in the sections as requested in order to facilitate reading to the reader and provide the manuscript with greater quality.
English in the manuscript is mostly good, but some parts of the text should be reviewed since there are some very long sentences with a muddle structure, especially in the introduction section. The editorial office will be able to assist you with the few corrections necessary when required.
The authors appreciate your comment. The introduction section has been revised with the aim of improving the understanding of the document and the quality of the manuscript.
Below I will make some specific comments on each section that should be carefully reviewed.
Introduction
The introduction is a bit general; it should be more focus on the aim of the work instead of in the general aspect of the sport physiology. In this sense, authors aim to characterize physical and physiological demands on different ages and genders in basketball, but there is nothing related to age or gender in basketball in the introduction that makes the reader understand why It is important to do this research.
The team of authors appreciate your comment with the aim of improving the quality of the manuscript. The introduction has been improved and modified in some of its sections.
In addition, it is necessary to add hypothesis to know what the authors expected to found with this research.
The research team has added the following hypothesis to the paper: “The hypothesis of this manuscript is that aerobic and anaerobic capacities will obtain an increase in the analysed values as the athlete develops. For this reason, older players will have better results in these capacities than younger players. In addition, differences between players of the same age, but of different gender will be confirmed. Male players are the ones who will obtain the best results in these physical fitness tests.”
Methods
From my view point “variables” section should be after procedure section since the reader could understand when and why each variable has been measured. In this sense, I think that the best way to order the methods section is Design – Participants – Procedures - Instruments – Variables – Statistical Analysis.
Thank you very much for your proposal, the authors really appreciate the comment in order to improve the quality of the manuscript. The method has been modified and finally the proposed order has been followed.
A characterisation (age, height, weight…) of each group is needed in participants section.
We do appreciate the comment. Your proposal has been accepted and is part of the final manuscript. The anthropometric characteristics have been added in the participants section. “149 players belonging to teams of different ages (U´14, U´16 and U´18), male and female who belong to the same club and participate in the national championship (U´14 male: n=33, U´14 female: n=12, U´16 male: n=31, U´16 female: n=12, U´18 male: n=39, U´18 female: n=22) were evaluated.” Has been changed by “149 male and female players that belong to teams of different ages (U´14, U´16 and U´18) and that belong to the same club and participate in the national championship (U´14 male: n=33, Weight = 62,20 kgs, Height= 1,72 metres, BMI= 20,78; U´14 female: n=12, Weight = 53 kgs, Height= 1,60 metres, BMI= 21,875; U´16 male: n=31, Weight = 76,81 kgs, Height= 1,87 metres, BMI= 21,91; U´16 female: n=12, Weight = 60,39 kgs, Height= 1,64 metres, BMI= 22,34; U´18 male: n=39, Weight = 85,23 kgs, Height= 1,95 metres, BMI= 22,41; U´18 female: n=22, Weight = 57,3 kgs, Height= 1,68 metres, BMI= 20,59) were evaluated.”
Results
The presentation of the results is a bit dense; there are 4 very large tables with many numbers. Maybe the presentation of some results in graphs could help to show and interpret them more easily.
Thank you very much for your comment in the form of a proposal. The results have been modified in order to facilitate the reading and interpretation of the document. In the previous version there were 4 very confusing tables. At present, this section has been modified. Now table 1 and 2 have become figures. The tables with the results of the figures have become part of Supplementary Materials. We trust it meets your expectations.
Discussion
As I point at the beginning of the review, discussion section is a bit long and at some point is hard to follow the text. Authors should consider reorganizing this section dividing results in two parts grouping some variables. They can do it in two different ways: age and gender or aerobic and anaerobic capacity.
The research team appreciates your proposal and attends to your comment. The discussion has been entirety modified order to facilitate reading. The discussion has been organised in two main sections (Aerobic Capacity and Anaerobic Capacity). In turn, each section has had several subsections (each group of variables). We trust it meets your expectations.
The second paragraph (Page 11 Lines 272-281) should be in the introduction section because it is a review of the state of art in relation with young athletes.
Thanks very much for your comment in the form of a proposal that provides us with a possible improvement of the document. The authors appreciate your comment and accept your proposal to change the paragraph to the Introduction.
In many parts of the discussion, authors explained the differences of some of the variables because of the level of maturation or the anthropometry, but there is not a characterisation of the sample in the methods section that justify the differences of the participants of each group.
Thank you very much for your comment and proposal, the anthropometric characteristics have already been added to Participants in the Method section.
Limitations and further research should be added at the end of the discussion section.
We appreciate your proposal. We have added the following to the manuscript: “The limitations found in this investigation were that the athletes analysed were part of teams that participated in the National Championship in their corresponding categories. This sample doesn´t provide a generic knowledge of the categories analysed since not all teams are part of this championship (only the best ones from different geographical regions). Besides, within the championship there are many differences between the teams that play the final stage of the championship and the teams that only play the group stage. Therefore, it would be interesting in further studies to expand the sample to teams from different geographical areas and with different results in national championships.”
Reviewer 2 Report
This study investigated 149 basketball players at different maturation level and both genders and aimed to characterize the physical-physiological demands of athletes in an aerobic and anaerobic test specific to basketball players, as well as the evolution of the variables according to age and gender. The findings are to be expected and merely discribe things that are more or less known. I accept that such studies are needed, however I personally do not find such studies interesting.
Furthermore, I think this study does not fall within the field of environmental research and public health and therefore not suitable for this journal. I think a sports science journal or a journal for coaches would be more suitable. Even though the authors report in great detail what they did and found out, I do not think their findings are of great interest to a wide range of readers. Finally, the language and presentation of the paper needs substantial improvement.
Author Response
Dear Editor,
We have carefully considered all reviewers' considerations of the paper (IJERPH-709605). Please find enclosed our detailed answers to reviewers' queries. Authors declare that the manuscript is original and has not been considered for publication elsewhere. Additionally, the authors had approved the paper for release and are in agreement with its content.
(x) I would not like to sign my review report.
() I would like to sign my review report English language and style.
(x) Extensive editing of English language and style required.
Thank you very much, the authors appreciate your comment in order to improve the quality of the manuscript and to facilitate reading to the reader. The research team has made different changes to the manuscript.
() Moderate English changes required.
() English language and style are fine/minor spell check required.
() I don't feel qualified to judge about the English language and style.
|
Yes |
Can be improved |
Must be improved |
Not applicable |
|
|
Does the introduction provide sufficient background and include all relevant references? |
() |
(x) |
( ) |
( ) |
|
Is the research design appropriate? |
() |
( ) |
(x) |
( ) |
|
Are the methods adequately described? |
() |
(x) |
( ) |
( ) |
|
Are the results clearly presented? |
() |
( ) |
(x) |
( ) |
|
Are the conclusions supported by the results? |
() |
(x) |
( ) |
( ) |
Comments and Suggestions for Authors
Does the introduction provide sufficient background and include all relevant references?
The authors appreciate your comment, the manuscript has been improved with the help of the comment proposed.
This study investigated 149 basketball players at different maturation level and both genders and aimed to characterize the physical-physiological demands of athletes in an aerobic and anaerobic test specific to basketball players, as well as the evolution of the variables according to age and gender. The findings are to be expected and merely discribe things that are more or less known. I accept that such studies are needed, however I personally do not find such studies interesting.
Furthermore, I think this study does not fall within the field of environmental research and public health and therefore not suitable for this journal. I think a sports science journal or a journal for coaches would be more suitable. Even though the authors report in great detail what they did and found out, I do not think their findings are of great interest to a wide range of readers. Finally, the language and presentation of the paper needs substantial improvement.
First of all, the authors really appreciate your comments on the manuscript. With regard to the sending of the document, it is true that the theme of the manuscript does not go along the same lines as the journal. However, the manuscript has been sent to this journal because there is a monograph entitled “Youth Sports, Young Athletes Evaluation, Implications for Performance and Health”. For this reason, since the manuscript evaluates capacities related to basketball performance in young players, the authors believe that it may be an appropriate part of this monograph. In addition, the language, the presentation of the results as well as the discussion have been modified with the sole objective of improving the quality of the manuscript.
Round 2
Reviewer 1 Report
From my viewpoint the manuscript has improved in relation with the previous version, especially with the presentation of the results in graphs instead of in such large tables. Nevertheless, there are still some issues that should be reviewed thoroughly, mainly in the discussion section.
Page 3 line 132: data collection instead of collection of data
Page 3 line 135: did the tests were randomized?
Page 3 line 141: add information about the video camera used (brand, frames/second…)
Page 4 line 147: add information about WIMU recordings (Hz)
Figures 1 and 2 are very pixelated. Authors should improve their resolution to be acceptable for the manuscript.
Authors did not understand what is expected with the discussion revision. It is still too long and they do no change almost anything but the order of the paragraphs. From my viewpoint, in this case, it is not necessary to discuss each variable isolated, since there are many of them and some of them with large paragraphs. They need to do an extra work by combining adequately different variables and giving a holistic approach to each aerobic and anaerobic capacity. A good example about what is expected is the last paragraph of the discussion section, where the authors made a final overview of gender characteristics in basketball aerobic and anaerobic capacities.
Moreover, there are duplicate information both in aerobic and anaerobic capacity sections; this is not admissible for a high quality publication. Some examples for this are:
The whole paragraph of Technical-Tactical variables (page 9 -10 lines 283-302 & page 11 – 12 lines 387 – 406) Kinematic variables re4lated to Accelerometry (page 10 lines 334 -337 & page 12 lines 436 – 439)Limitations and further research sections should be addressed at the end of discussion.
Author Response
From my viewpoint the manuscript has improved in relation with the previous version, especially with the presentation of the results in graphs instead of in such large tables. Nevertheless, there are still some issues that should be reviewed thoroughly, mainly in the discussion section.
Page 3 line 132: data collection instead of collection of data
The research team appreciates the proposed change in the error. It has been modified being as you propose.
Page 3 line 135: did the tests were randomized?
Thank you very much for the improvement proposal. The order or the test has been detailed in the manuscript: “The first day the aerobic capacity test. On the second day, the lactic anaerobic capacity test was performed”
Page 3 line 141: add information about the video camera used (brand, frames/second…)
Thank you very much for the proposal. The information related to the video camera has been added to the manuscript. It has been added: “JVC model GY-HM70U with a sampling rate of 300 fps (resolution 720x480)."
Page 4 line 147: add information about WIMU recordings (Hz)
The WIMU inertial device has a sampling frequency of 18 Hz. This information has been added to the manuscript: “each player was equipped with a WIMUÒ inertial device from RealTrack Systems (Almería, Spain), which was fixed using a harness anatomically adjusted to each player. After registration, the data was analysed using the SPROÒ software from RealTrack Systems (Almería, Spain). The WIMU inertial device has a sampling frequency of 18 Hz”.
Figures 1 and 2 are very pixelated. Authors should improve their resolution to be acceptable for the manuscript.
The research team apologizes for the quality of the figures. They have been modified and improved so that the quality is optimal. The authors hope you like the quality of the image.
Authors did not understand what is expected with the discussion revision. It is still too long and they do no change almost anything but the order of the paragraphs. From my viewpoint, in this case, it is not necessary to discuss each variable isolated, since there are many of them and some of them with large paragraphs. They need to do an extra work by combining adequately different variables and giving a holistic approach to each aerobic and anaerobic capacity. A good example about what is expected is the last paragraph of the discussion section, where the authors made a final overview of gender characteristics in basketball aerobic and anaerobic capacities.
The research team apologizes for the discussion by not correctly understanding what it´s requesting. The revision has been modified again with the objetive of fulfilling what is requested. The size has been reduced and it has been specified in relevant information. The authors hope you like it. If on the contrary we did not, we are completely at your disposal to make the changes that create opportunities.
Moreover, there are duplicate information both in aerobic and anaerobic capacity sections; this is not admissible for a high quality publication. Some examples for this are:
The whole paragraph of Technical-Tactical variables (page 9 -10 lines 283-302 & page 11 – 12 lines 387 – 406) Kinematic variables re4lated to Accelerometry (page 10 lines 334 -337 & page 12 lines 436 – 439)
The research team apologizes for the duplication of information errors. The discussion has been modified and these errors no longer exist.
Limitations and further research sections should be addressed at the end of discussion
The limitations were at the end of the conclusions, but according to your proposal, the have been changed at the end of the discussion section.

Reviewer 2 Report
This manuscript has substantially improved after this revision. Congratulations to the authors.
I was not aware of this special issue on the topic where this paper fits in very well indeed.
I personally do not find this study interesting and it also lacks novelty. However, I accept that such studies are needed and this one is well conducted and the authors did a great job in how to adequately report their findings now.
Author Response
This manuscript has substantially improved after this revision. Congratulations to the authors.
The research team appreciates your words. This improvement has been partly thanks to the work as a reviewer.
I was not aware of this special issue on the topic where this paper fits in very well indeed.
I personally do not find this study interesting and it also lacks novelty. However, I accept that such studies are needed and this one is well conducted and the authors did a great job in how to adequately report their findings now.
The research team appreciates your words. The novelty of this research is not the subject, since there are many investigation with similar themes. The novelty of this manuscript and what it makes different is the material used (inertial devices WIMU) and the instruments used (test). The tests used in the research are those chosen by the research team because they are the most specific tests that exist to assess aerobic and anaerobic capacities on the playing field (outside the laboratory).

Round 3
Reviewer 1 Report
The authors have taken into account all the comments in the latest version submited and the quality of the manuscript has increased significantly.